# Attitudes about Mechanical Restraint Use in Mental Health Hospitalization Services: A Spanish Survey

**DOI:** 10.3390/healthcare11131909

**Published:** 2023-07-01

**Authors:** Carlos Aguilera-Serrano, Jessica Marian Goodman-Casanova, Antonio Bordallo-Aragón, Juan Antonio García-Sánchez, Fermín Mayoral-Cleries, José Guzmán-Parra

**Affiliations:** 1Unidad de Gestión Clínica de Salud Mental, Hospital Regional Universitario de Málaga, Instituto de Investigación Biomédica de Málaga y Plataforma en Nanomedicina (IBIMA Plataforma Bionand), 29009 Málaga, Spain; c_aguiler2@hotmail.com (C.A.-S.); antonio.bordallo.sspa@juntadeandalucia.es (A.B.-A.); juan.garcia.sanchez.sspa@juntadeandalucia.es (J.A.G.-S.); fermin.mayoral.sspa@juntadeandalucia.es (F.M.-C.); 2COST Action CA19133–Fostering and Strengthening Approaches to Reducing Coercion in European Mental Health Services (FOSTREN), 1050 Brussels, Belgium

**Keywords:** mechanical restraint, coercion, safety and security, service evaluation, staff

## Abstract

The aim of this study was to analyze the attitudes of professionals in Mental Health Services throughout Spain who are directly or indirectly involved in the use of mechanical restraint and the barriers perceived to reduce its use. The study involved an online anonymous survey using Google Forms completed by Spanish mental health professionals working with service users; the survey assessed their involvement in and general attitudes and beliefs towards mechanical restraint. The survey was completed by 225 participants. Only 13.30% of the participants considered that mechanical restraint use was never necessary to guarantee the safety of users/staff in dangerous situations. Poor staff training (38.0%) and a lack of resources/staff (34.7%) were the most frequent barriers identified for the reduction of mechanical restraint. In the multivariate analysis, participation in learning programs to prevent the use of mechanical restraint was associated with lower acceptance of the use of mechanical restraint, but the result was barely significant (*p* = 0.050). A high percentage of mental health staff still consider mechanical restraint use necessary for safety reasons. According to the results, the participants perceived that more staff and resources and better training could reduce the use of mechanical restraint in Mental Health Hospitalization Services.

## 1. Introduction

The use of mechanical restraint and other coercive measures in Mental Health Hospitalization Services in Spain remains very frequent [1]. However, there is scarce evidence that supports its usefulness [2] and plenty of evidence that reductions in the use of coercive measures are not associated with increases in undesired outcomes. Mechanical restraint has been associated with negative physical, psychological, and/or cognitive consequences, and with negative impacts on the therapist–patient relationship [3]. In addition, this practice has been associated with lower satisfaction with hospitalization [4], increased exposure of healthcare professionals to occupational hazards [5] and psychological burdens [6,7], and significant economic costs for the administration—as pointed out by the data from a systematic review, estimating a total annual cost of EUR 28,518 per ward [8]. Moreover, user associations actively demand its elimination [9] and replacement by more humanized services, demanding compliance with international legislation to promote and protect the human rights and dignity of people with disabilities [10]. However, mechanical restraint and other coercive measures are mainly used as a consequence of violence towards staff, family, other users, or self-harm [11,12], and some staff and users perceive these practices as positive and necessary to maintain safety [3]. Additionally, it has been argued that the beneficence and human dignity argument is not sufficient to reject coercive measures in mental health services [13].

Although there are some initiatives at the national level to reduce this practice [14,15], many hospitalization services have not been able to reduce its use [16]. This situation is also shown by international statistics—as revealed by the European Evaluation of Coercion in Psychiatry and Harmonization of Best Clinical Practices [16], which evaluated and compared the use of restrictive measures in psychiatric hospitalization services in ten European countries, with results ranging from 15% to 55% (37% in Spain) [16].

Among the factors that contribute to the difficulty of eliminating or reducing this coercive measure are the attitudes and beliefs of healthcare professionals, organizational barriers, and the shortage of resources for implementing alternatives [17]. Different international studies show that healthcare professionals responsible for implementing these measures are the group that perceives them as the most necessary to guarantee user safety [16]. Likewise, it is a practice associated with a lack of resources [12], which prevents the implementation of more complex actions that require better trained and more numerous personnel. However, the practice of mechanical restraint and coercive measures is very different in each country, and even between different regions [18,19]. Little is still known about the attitudes and perceived barriers to reducing this practice by healthcare professionals, and there is no comprehensive study in Spain on the attitudes and beliefs towards this practice in healthcare staff from Mental Health Hospitalization Services involved in its use [20].

The Spanish health system depends on the systems of each of the seventeen autonomous communities, and there is not a unified protocol for the application of coercive measures and mechanical restraint. In general, the ultimate responsibility of the procedure of mechanical restraint lies with the psychiatrist and the nurses, as the staff responsible for implementing the procedure and supervising the safety of the user during restraint.

The aim of this study was to analyze the attitudes of healthcare professionals from Mental Health Services throughout the Spanish territory that are directly or indirectly involved in the use of mechanical restraint, and the perceived barriers in reducing its use. Additional aims were to analyze if there were differences between nurses and other staff in terms of attitudes regarding mechanical restraint.

## 2. Materials and Methods

### 2.1. Design

A cross-sectional observational study was carried out. The study involved delivering an online, anonymous survey to Spanish mental health professionals working with mental health users using Google Forms. The survey was used to investigate their attitudes and beliefs regarding the use of mechanical restraint. The definition of this practice in the survey was as follows: “Mechanical restraint is defined as any manual/physical method or mechanical device, material, or equipment that immobilizes or reduces the ability of a person to move his or her arms, legs, body, or head freely” [21].

The respondents recruited were members of the Spanish Association of Mental Health Nursing (AEESME), the Spanish Association of Neuropsychiatry (AEN), the Spanish Association of Clinical Psychology and Psychopathology (AEPCP), and different technical advisors of Mental Health Regional Programs. Details of the research project were made available to members of these groups through their email distribution lists (for AEESME, AEN, and AEPCP members), websites, social media platforms (e.g., Twitter and Facebook pages), and newsletters. Information provided to members consisted of a short description of the project and the URL to access the survey. The survey was available from 1 May to 1 September 2022.

### 2.2. Ethical Considerations

The Provincial Ethics Committee of Malaga authorized the study (Code: CoerSt-01). An information letter for the study was given to participants in the cover section of the questionnaire to explain the study’s purpose, procedure, voluntary participation, informed consent, and privacy issues. Anonymity and confidentiality were ensured by anonymizing all questionnaires.

### 2.3. Participants

The survey was completed by 225 participants: mean age 41.48 years, 71.60% female, 65.33% nurses, and 56.00% involved in mechanical restraint use in the last year. More information about the characteristics of the participants is shown in Table 1.

### 2.4. Measures

Sociodemographic and staff-related variables included in the study were: age, sex, service where the participant worked, type of staff (nurse, other professionals) and years of experience. Additionally, variables related to the practice of mechanical restraint were collected: use of mechanical restraint ever (yes/no), use of mechanical restraint in the last year (yes/no), existence of updated protocols about the use of mechanical restraint (yes/no), as well as whether they had received any type of training in the last year (yes/no).

To measure the attitudes and beliefs of mechanical restraint use, items in the survey were either drawn from previously designed measures of attitudes to restraint and working practices with psychiatric patients or specifically written for the project. Items were completed using a 5-point Likert-type response scale, ranging from 0 (strongly disagree) to 4 (strongly agree) for items exploring the professionals’ attitudes; and ranging from 0 (very unlikely) to 4 (very likely) for items measuring the likelihood of mechanical restraint use.

Attitudes and beliefs towards mechanical restraint use were examined using specifically written items based on the literature [22] and items adapted from three measures. The Staff Attitude to Coercion Scale (SACS) [23] measures nurses’ perceptions regarding seclusion and restraint use, including the extent to which these practices prevent dangerous situations, are necessary, and can be reduced. The Seclusion and Restraint Experience Questionnaire (SREQ) [24] measures nurses’ emotions towards and experiences of the use of seclusion/restraint, and perceptions of the ethical/practical implications of their use. Finally, the Attitudes to Containment Methods Questionnaire (ACMQ) [25] encompasses items relating to restraint methods that are used in European psychiatric care. This section comprised 19 items (SACS: items 5, 6, 8, 10, 13 and 16; SREQ: items 9, 11, 12, 14, 15, 17 and 19; ACMQ: items 1, 2, 3, 4, 7 and 18. The number of the items is shown in Table 2, Table 3 and Table 4) and examined professional attitudes toward mechanical restraint use, including: (1) perceived acceptability, (2) perceived negative consequences, and (3) possible prevention strategies. In addition, two short-response items were included to allow respondents to answer openly regarding the main important negative consequences and barriers to reducing mechanical restraint use.

### 2.5. Data Analysis

A descriptive analysis, using the mean and the standard deviation, was carried out considering all the items of the survey. Regarding the two open questions of the survey, a content analysis was carried out and frequency and percentages used. We carried out a multivariate linear regression analysis to assess the factors associated with the perceived acceptability of the use of mechanical restraint. The dependent variable was a score calculated by summing all items related to perceived acceptability. To verify the assumptions of the linear regression model, we used the Breusch–Pagan homoscedasticity test, the Shapiro–Wilks normality test of the residuals, and a scatter plot to verify the linear relationship of the variables. The variance inflation factor was used to evaluate the possible collinearity of the variables. In the multivariate model, the following variables were introduced: age, sex, service where the participant worked, type of staff (nurse, other professionals), years of experience, use of mechanical restraint ever (yes/no), existence of updated protocols concerning the use of mechanical restraint (yes/no), as well as whether they had received any type of training in the last year (yes/no). For the comparison between nurses and other professional groups, we used the chi-squared test for categorical variables and the Mann–Whitney U test for quantitative variables. For all comparisons, an alpha level of 0.05 was used. We used R program version 4.2.2 (R Foundation for Statistical Computing, Vienna, Austria) and the R Commander package for the analysis.

## 3. Results

### 3.1. Perceived Acceptability

Of the respondents, 21.33% of the sample strongly agreed with this statement that mechanical restraint use is necessary for the safety of users/staff in dangerous situations and only 13.30% of the participants strongly disagreed with the statement. This statement had the highest level of agreement (M = 2.22, SD = 1.34), followed by “I believe that the professional legal framework of my discipline supports the application of mechanical restraint” (M = 2.16, SD = 1.27). The statement with the highest level of disagreement was “the use of mechanical restraint sets limits to the users“ (M = 1.04, SD = 1.07) and “it is important to apply mechanical restraint to guarantee the legal protection of healthcare professionals and the hospital itself “ (M = 1.09, SD = 1.15). More information about the acceptability of mechanical restraint use is shown in Table 2.

### 3.2. Perceived Negative Consequences

Of the participants, 60% strongly agreed that mechanical restraint use could damage the therapeutic relationship, and only 5.33% completely disagreed—being the item with the most agreement (M = 3.28, SD = 1.11). The item with the second highest level of agreement was “A user suffers a loss of dignity when he/she is subjected to mechanical restraint“ (M = 2.94, SD = 1.26). More disagreement was shown in the item “I feel that placing a user in mechanical restraint can reduce nursing care and attention time” (M = 1.89, SD = 1.49). A21.33% of the participants strongly agreed that mechanical restraint use represents a failure of the healthcare team, while 16.44% strongly disagreed with this assumption—being the statement with second lowest mean score of agreement (M = 2.13, SD = 1.37). In the open question regarding the consequences of mechanical restraint, 39.11% indicated negative emotional consequences, 18.66% indicated damage to the therapeutic relationship or loss of confidence in staff, 17.77% indicated physical negative consequences, 8.44% indicated a violation of human rights, and 1.77% indicated other negative consequences. However, 8.44% of participants indicated positive consequences of mechanical restraint, and 5.88% did not respond to the question. More information about perceived negative consequences is shown in Table 3.

### 3.3. Perceived Possibilities for Prevention

Most agreement was shown regarding the use of alternative methods to mechanical restraint (M = 3.80, SD= 0.79) and with the statement that the “most important thing is to inform the user that you care about him or her” (M = 3.44, SD = 0.94). The most disagreement was shown regarding whether “alternative methods could not fully replace the use of mechanical restraint”: 11.11% strongly agreed and 26.22% strongly disagreed (M = 1.55, SD = 1.32). The second was that “it is difficult to find and implement alternative methods in the work environment” (M = 1.61, SD = 1.27). Regarding the open question, the most frequent main barrier identified for the reduction of mechanical restraint use was poor staff training (39.0%), followed by a lack of resources/staff (34.7%) as the second reason. The third reason was related to culture, organizational factors, and the mentality of staff (18.33%). More information concerning perceived barriers for the reduction of mechanical restraint use is shown in Table 4.

### 3.4. Factors Associated with Acceptability of the Use of Mechanical Restraint

In the multivariate analysis, participation in learning programs for the prevention of the use of mechanical restraint was negatively associated with the acceptability of its use, but the result was only marginally significant (B = −1.569; *p* = 0.050). More information concerning the multivariate model is shown in Table 5.

### 3.5. Differences between Nurses and Other Staff

Nurses considered mechanical restraint use to increase the safety of users (*p* = 0.034) and staff (*p* = 0.012) more than other staff. Regarding the perceived negative consequences of mechanical restraint, there was no statistically significant difference between nurses and other professionals regarding any of the items. Nurses thought more often than other professionals that the most important thing was to inform the user that they care about him or her (*p* = 0.007), and less often thought that it was difficult to find and implement alternative methods in their work environment (*p* = 0.035). More information about the differences between nurses and other staff is shown in Table 2, Table 3 and Table 4.

## 4. Discussion

This is the largest study carried out in Spain to date on the attitudes and beliefs on mechanical restraint use among healthcare mental health professionals. Overall, respondents believed that the complete elimination of mechanical restraint use is not possible, as they perceived it to be necessary in extreme situations to protect users and staff. Additionally, as in other studies, staff members who physically participated in mechanical restraint were significantly more likely to agree with statements indicating that this measure is a means to achieving safety, care, and order [2,26]. The survey identified opinions from staff concerning different measures that would likely help or hinder efforts to reduce the use of restraint.

The judgments concerning the acceptability of mechanical restraint use by the respondents were consistent with those of other studies—particularly if we focus on the majority group of mental health nurses, who considered this measure to be riskier and/or more harmful to the user in general [12]. However, it seems that nurses who had received prior training on how to prevent the use of mechanical restraint reported lower levels of intention to use restraint. Studies conducted in social and cultural contexts similar to our sample indicate the key role that nurses play in detecting difficult situations, giving them the power to prevent the use of mechanical restraint [27]. Although the study participants stated that the attitudes of team members were often decisive in the choice to use mechanical restraint, different attitudes in the team and different ways of dealing with restraint were rarely addressed.

Most of the respondents highlighted that the implementation of mechanical restraint is detrimental to the therapeutic interpersonal relationship, which is essential for providing quality care and promoting recovery [7,28]. Mechanical restraint can also elicit feelings of guilt and perceptions of malpractice among the professionals themselves. Previous studies have reported that ethical reflection among healthcare providers is key to preserving empathy, and that empathy among care staff is crucial for reducing the use of restraints [7,29]. Increasing satisfaction with psychiatric treatment is becoming more important over time and can influence the therapeutic relationship and the long-term effectiveness of treatment [30].

One widely reported aspect in the available evidence is the widespread perception of the loss of dignity suffered by the person subjected to restraint—something that was reflected by the respondents. The principles of “use of force” inherent in this practice likely carry a high risk of harm and interference with human dignity [31].

In relation to perceived possibilities for the prevention and reduction of the use of mechanical restraint, the majority of respondents stated that effective actions can be implemented. According to the reviewed literature, it is possible to reduce the use of mechanical restraints and coercive measures and not increase the number of incidents and violent behaviors among users through a non-invasive and non-pharmacological approach. Training in de-escalation techniques, risk assessment, and implementation of the “six core strategies” or “Safewards” program were the most evaluated and effective interventions in reducing aggressive behaviors and the use of coercive measures [32]. Multiple studies have found that proper treatment by staff, providing information, respect, and subsequent questioning [33] influence the perception of coercion—explaining their relationship to satisfaction with treatment—and indicates that efforts to improve professional procedures can decrease perceived coercion and increase satisfaction with treatment. However, there was also a high degree of agreement that mechanical restraint is necessary to some extent to maintain the safety of hospital units, which demonstrates that a significant proportion of staff believe that the complete elimination of this measure would be unfeasible—aligning with the perception shown in other international studies [12,18]. This indicates that efforts should continue towards developing better interventions and safer environments that provide a greater sense of security, allowing further reductions in and, if possible, the complete elimination of mechanical restraint.

This study has some limitations that must be considered when interpreting the results. First, it is an online convenience survey in which most of the sample comes from Andalusia. People willing to answer an online survey are likely to have specific characteristics that may introduce some biases—including that most of the sample comes from people who belong to associations that may have a higher commitment to improving professional practice. Additionally, the sample is from only one country, and international comparisons may not be appropriate. Another limitation of the study is that the sample of non-nurse professionals is diverse and not very large, which may limit it statistical power. Furthermore, since the use of mechanical restraint is a sensitive issue, social desirability bias may have played a relevant role.

## 5. Conclusions

An important percentage of mental health staff still consider mechanical restraint use as necessary in guaranteeing the safety of the users and staff, and a relevant percentage of participants still consider the main consequences of mechanical restraint as positive. Following the results of this study, professionals perceived that more staff and resources and better training could reduce the use of mechanical restraint in Mental Health Hospitalization Services. This knowledge concerning present attitudes regarding mechanical restraint in the workforce could be useful in designing future training and clarifying possible barriers that prevent its reduction. Future research must analyze if the attitudes of staff and their changes are relevant and if they influence the reduction of coercive measures and mechanical restraint.

## Figures and Tables

**Table 1 healthcare-11-01909-t001:** Characteristics of participants.

Characteristics of Participants	Total (%) n = 225	Nurses (%) n = 147	Other Professionals (%) n = 78	Statistic/*p* Values
Gender				9.289/**0.002** Chi square
Female	161 (71.6)	115 (78.2)	46 (59.0)	
Male	64 (28.4)	64 (21.8)	32 (41.0)	
Age (Mean, standard deviation)	41.48 (12.33)	40.19 (12.54)	43.90 (11.61)	−2.360/**0.018** Mann–Whitney
Years of experience				9.191/**0.010** Chi square
0–10	101 (44.9)	75 (51.0)	26 (33.3)	
11–20	54 (24.0)	27 (18.4)	27 (34.6)	
>20	70 (31.1)	45 (30.6)	25 (32.1)	
Mental Health Service				28.078/**<0.001** Chi square
Hospitalization	85 (37.8)	72 (49.0)	13 (16.7)	
Community	51 (22.7)	21 (14.3)	30 (38.5)	
Other	89 (39.6)	54 (36.7)	35 (44.9)	
Use of Mechanical restraint				4.944/**0.026** Chi square
Yes	206 (91.6)	139 (94.6)	67 (85.9)	
No	19 (8.4)	8 (5.4)	11 (14.1)	
Use of Mechanical restraint last year				3.554/0.059 Chi square
Yes	126 (56.0)	89 (60.5)	37 (47.7)	
No	99 (44.0)	58 (39.5)	41 (52.6)	
Presence of specific and updated protocol				3.752/0.053 Chi square
Yes	164 (72.9)	101 (68.7)	63 (80.8)	
No	61 (27.1)	46 (31.3)	15 (19.2)	
Participation in learning program to prevent the use of mechanical restraint				2.636/0.104 Chi square
Yes	154 (68.4)	106 (72.1)	48 (61.5)	
No	71 (31.6)	41 (27.9)	30 (38.5)	

**Table 2 healthcare-11-01909-t002:** Perceived acceptability of the use of mechanical restraint.

Characteristics of Participants	Total n = 225 Mean (SD)	Nurses n = 147 Mean (SD)	Other professionals n = 78 Mean (SD)	Stadistic/*p* Values	Strongly Disagree/Strongly Agree %
1. The use of mechanical restraint is necessary for protection in dangerous situations (0–4 Score)	2.22 (1.34)	2.21 (1.31)	2.24 (1.40)	−0.226/0.821 Mann–Whitney	13.3/21.3
2. The use of mechanical restraint increases the safety of the user (0–4 Score)	1.31 (0.95)	1.23 (0.95)	1.47 (0.94)	−2.118/**0.034** Mann–Whitney	15.1/3.6
3. The use of mechanical restraint increases the safety of the staff (0–4 Score)	1.65 (0.98)	1.54 (0.95)	1.87 (1.00)	−2.509/**0.012** Mann–Whitney	5.8/4.4
4. The use of mechanical restraint sets limits to the users (0–4 Score)	1.04 (1.07)	0.94 (0.99)	1.23 (1.18)	−1.614/0.107 Mann–Whitney	36.9/2.2
5. Mechanical restraint can represent care and protection (0–4 Score)	1.85 (1.30)	1.84 (1.32)	1.86 (1.28)	−0.159/0.874 Mann–Whitney	20.0/12.9
6. It is important to apply mechanical restraint to guarantee the legal protection of healthcare professionals and the hospital itself (0–4 Score)	1.09 (1.15)	1.12 (1.17)	1.04 (1.11)	−0.626/0.531 Mann–Whitney	40.4/4.4
7. I believe that the professional legal framework of my discipline supports the application of mechanical restraint (0–4 Score)	2.16 (1.27)	2.12 (1.24)	2.23 (1.31)	−0.431/0.667 Mann–Whitney	13.8/16.4

Note: All items were measured on a Likert scale from 0 to 4.

**Table 3 healthcare-11-01909-t003:** Perceived negative consequences of the use of mechanical restraint.

Characteristics of Participants	Total n = 225 Mean (SD)	Nurses n = 147 Mean (SD)	Other Professionals n = 78 Mean (SD)	Stadistic/*p* Values	Strongly Disagree/Strongly Agree %
8. Mechanical restraint can damage the therapeutic relationship (0–4 Score)	3.28 (1.11)	3.22 (1.14)	3.38 (1.05)	−1.117/0.241 Mann–Whitney	5.3/60.0
9. I feel ashamed when I inform the family of the user that he/she is restrained (0–4 Score)	2.85 (1.25)	2.84 (1.31)	2.85 (1.15)	−0.374/0.709 Mann–Whitney	8.4/41.6
10. The use of mechanical restraint is a failure by the healthcare team (0–4 Score)	2.13 (1.37)	2.14 (1.35)	2.12 (1.41)	−0.121/0.904 Mann–Whitney	16.4/21.3
11. I feel guilty applying mechanical restraint (0–4 Score)	2.34 (1.42)	2.39 (1.38)	2.24 (1.50)	−0.655/0.513 Mann–Whitney	14.9/27.9
12. It makes me feel bad if the user gets angrier after applying mechanical restraint (0–4 Score)	2.45 (1.34)	2.50 (1.29)	2.35 (1.42)	−0.626/0.531 Mann–Whitney	11.6/27.4
13. I feel that placing a user in mechanical restraint can reduce nursing care and attention time. (0–4 Score)	1.89 (1.49)	1.78 (1.53)	2.12 (1.39)	−1.619/0.105 Mann–Whitney	27.6/19.6
14. A user suffers a loss of dignity when he/she is subjected to mechanical restraint. (0–4 Score)	2.94 (1.26)	2.97 (1.18)	2.90 (1.39)	−0.232/0.816 Mann–Whitney	7.1/45.8

Note: All items were measured on a Likert scale from 0 to 4.

**Table 4 healthcare-11-01909-t004:** Perceived possibilities to the prevention and reduction of the use of mechanical restraint.

Characteristics of Participants	Total n = 225 Mean (SD)	Nurses n = 147 Mean (SD)	Other Professionals n = 78 Mean (SD)	Stadistic/*p* Values	Strongly Disagree/Strongly Agree %
15. I always try (or would try) alternative measures before mechanically restraining a user (0–4 Score)	3.80 (0.79)	3.82 (0.75)	3.74 (0.86)	−0.734/0.463 Mann–Whitney	3.6/91.6
16. Alternative methods cannot fully replace the use of mechanical restraint (0–4 Score)	1.55 (1.32)	1.43 (1.29)	1.78 (1.36)	−1.879/0.060 Mann–Whitney	26.2/11.1
17. It is difficult to find and implement alternative methods in my work environment (0–4 Score)	1.61 (1.27)	1.48 (1.25)	1.85 (1.26)	−2.103/**0.035** Mann–Whitney	22.2/9.8
18. In critical situations, I think the most important thing is to inform the user that I care about him or her. (0–4 Score)	3.44 (0.94)	3.56 (0.83)	3.22 (1.10)	−2.701/**0.007** Mann–Whitney	3.6/64.4
19. It is difficult to decide when to correctly indicate mechanical restraint. (0–4 Score)	2.60 (1.31)	2.60 (1.31)	2.62 (1.31)	−0.096/0.924 Mann–Whitney	9.3/31.6

Note: All items were measured on a Likert scale from 0 to 4.

**Table 5 healthcare-11-01909-t005:** Factors associated with acceptability of the use of mechanical restraint.

Dependent Variable = Acceptability Score ^1^					
Variables	Coeficients	Error	t	*p*	VIF *
Intercept	8.191	2.485	3.297	0.001	
Sex					1.101
Female	Ref				
Male	−0.074	0.804	−0.089	0.929	
Mental Health Service				0.105	1.192
Hospitalization	Ref				
Community	−1.345	1.032	−1.303	0.193	
Other	−1.785	0.847	−2.108	**0.036**	
Type of Staff					1.294
Other	Ref				
Nurse	−1.248	3.500	−0.143	0.887	
Age	0.104	0.059	1.768	0.078	4.028
Use of Mechanical restraint					1.092
No	Ref				
Yes	1.992	1.349	1.477	0.141	
Presence of specific and updated protocol					1.065
No	Ref				
Yes	1.370	0.833	1.644	0.102	
Participation in learning program to prevent use of mechanical restraint					1.063
No	Ref				
Yes	−1.569	0.796	−1.971	0.050	
Years of experience				0.214	4.199
0–10	Ref				
11–20	−2.087	1.192	−1.751	0.081	
>20	−1.775	1.660	−1.070	0.285	

^1^ R^2^ = 0.084. Breusch–Pagan Homoscedasticity test = 0.410. Shapiro–Wilks test of residuals = 0.052. * Note: VIF >5 indicates that the variable is highly correlated.

## Data Availability

The dataset may be available under reasonable request. Requests to access these datasets should be directed to JG-P; jose.guzman.parra.sspa@juntadeandalucia.es.

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
