# Peer review of "Attitudes about Mechanical Restraint Use in Mental Health Hospitalization Services: A Spanish Survey"

_healthcare, 2023, doi:10.3390/healthcare11131909_

Round 1
Reviewer 1 Report
The authors describe and analyze mental health services professionals´ attitudes towards the use of mechanical restraint by a survey.
Abstract: The authors did not mention that the comparation of the attitudes of single professional groups (nurses vs. other professionals) was one of the main goals of the study.
Introduction: The presentation of the professional basis of physical restraints remains one-sided. This creates the impression that there is a "right" and a "wrong" opinion in the subsequent survey, even though the experts are engaged in an intensive discourse around this topic. A more objective approach would be better here, showing why physical restraints are still used today. Even if the perspective of the user associations was presented correctly, the protection of all those involved is out of the focus. I would find it more appropriate to reflect on the law, in which the protection of third parties usually also plays a role. E. g., Line 52 formulates mechanical restraint as a necessary evil, this too tendentious
Aim: „analyse the attitudes if healthcare professionals directly or indirectly involved“ - Is this tied to professional groups? Are only nurses responsible for mechanical restraint in Spain? Please clarify. The authors assume the knowledge of the legal situation in Spain, for an international publication an explanation of the legal procederes is absolutely necessary.
Design: a well written chapter, measures are presented transparently, the combination of different questionnaires is well chosen
Data analysis: The comparison of the two professional groups is very dominant and disproportionate in the focus of the analysis. The actual topic and aim of the study is not focused.
Results: It gives the appearance that the results initially presented in the text are also part of the illustrations, which is not the case. The %-values in the tables 2-4 are not understandable. Potentially, the illustrations present the means of the Likert scales and their standard deviation, which are the compared between the groups. This should be clearly stated in the headline.
Again: the comparison between the groups takes up a lot of space. If this is the primary question, it should be specified earlier and more precisely.
Discussion:
Even with the attitude, that physical restraints in psychiatry is used too often, too long and could be avoided in several or many cases, the discussion represents no objective point of view.
Line 188 - "eliminate the use of restraint“ - I think this is uncritical, one-sided and a not achievable goal. Also, it should be highlighted that these are subjective assessments of employees- whether and to what extent frequencies of mechanical restraint can actually be reduced is part of ongoing research and cannot be quantified precisely.
Line 195: "giving them the power to prevent the use of mechanical restraint“- This sounds too strong to me, because formally, they do not have much power, but they certainly have influence.
Line 202: "attempting to control the user through mechanical restraint can reduce the time for care and attention from professionals" - that may in fact be true, but this problem won’t be solved by waiving to mechanical restraint, but by more staff.
Line 220: Just a comment: even clinics that use Safewards etc. cannot work without coercive measures.
Minor points:
Introduction 1st paragraph: Moreover, user associations demand actively it’s (spelling error) elimination in defense of more humanized services, demanding compliance with international legislation to promote and protect the human rights and dignity of people with disabilities
OK.
Reviewer 2 Report
This cross-sectional observational study aimed to identify the attitudes and beliefs of mental health healthcare professionals in Spain who work with mental health patients on using mechanical restraints and perceived barriers to reducing mechanical restraint use. The study's main contributions include the consistency of the findings with previous research (adding to the power) that healthcare professionals' attitudes and beliefs of the use of mechanical restraints are that it is harmful to patients and has minimal positive benefits. Barriers to reducing restraints continues to be an issue. Staff training may have a positive effect on reducing restraint use. The study's strengths were the use of the IMRAD format; an appropriate synthesis of the background information that outlined the importance of the problem, the cost of not addressing the problem to the healthcare system, the negative impact on patients, and the gap in the literature justifying the research study. The references were mostly within the last five years and appropriate to the topic of the study. The author identified a conceptual definition of mechanical restraints. The research design was appropriate for the purpose of the study. The study's protection of human subjects procedures was appropriate using anonymous surveys and obtaining the ethics committee's approval. The data analysis section identified the statistical software program used, the appropriate descriptive and statistical tests, and the assumptions. The results identified the statistically significant results and were tied to the purpose of the study. The discussion section linked the results to previous research and identified the study's limitations. The conclusion is consistent with the results.
Line 30-31 " there is no evidence that supports its usefulness" I recommend adding a citation.
In lines 41 & 42, there is a contradiction with the cited source. According to the cited source, some hospitals have reduced restraints. Consider rewording to discuss the lack of regional and national strategies to address mechanical restraints.
Lines 61 - 63 I recommend rewriting the aim to specifical include the analysis of healthcare professionals' attitudes and beliefs on mechanical restraints. The aim is unclear. Line 68 includes "perceived barriers" in reducing mechanical restraint use.
The methods sections: The authors discuss the different instruments used to develop the survey. Still, it is unclear how many items were included in the survey, which items from each survey were used, and any questions that were developed. The citations support the reliability and validity of the SACA and the SREQ but no report on the ACMQ. Include the example of the survey used in the study and report the current reliability of this study.
Line 135- Level of significance needs clarification. How it is written refers to a confidence level of 95%. The significance level (alpha level) of .05 matches the reported p-value of .05 in the tables and result section.
Table 1 Age is a continuous variable, but a Mann-Whitney test was used.
Include a symbol in the table key to identify statistically significant results in the tables.
For clarification, add a table to report the results of the perceived acceptability, a table to report the perceived negative consequences, and perceived possibilities for the prevention (part of the aim of the study). Then you can discuss the statistically significant results in the narrative.
In the key for Table 5, identify what indicates a good VIF.
Line 192-193 Need to report data in the result section to support this statement, "nurses who had received prior training on how to prevent the use of mechanical restraints reported lower levels of intention to use restraints."
Add in the discussion section future research and implications for practice.
Round 2
Reviewer 1 Report
The manuscript has clearly improved.